# Differentially Expressed miRNAs and mRNAs in Regenerated Scales of Rainbow Trout (*Oncorhynchus mykiss*) under Salinity Acclimation

**DOI:** 10.3390/ani12101265

**Published:** 2022-05-14

**Authors:** Changgeng Yang, Qiling Zhou, Qian Ma, Liuyong Wang, Yunsheng Yang, Gang Chen

**Affiliations:** 1Life Science & Technology School, Lingnan Normal University, Zhanjiang 524048, China; yangcg910@163.com; 2College of Fisheries, Guangdong Ocean University, Zhanjiang 524025, China; 18738627206@163.com (Q.Z.); wlyxkay@163.com (L.W.); 2112101049@stu.gdou.edu.cn (Y.Y.); cheng@gdou.edu.cn (G.C.); 3Southern Marine Science and Engineering Guangdong Laboratory, Zhanjiang 524025, China

**Keywords:** salmonid species, hypersalinity, ncRNA, omics, exoskeleton

## Abstract

**Simple Summary:**

To further understand the molecular mechanism underlying the bone remodeling process, transcriptional information of regenerated scales of rainbow trout under sea water (SW) and fresh water (FW) environments was compared in this study. The miRNA and mRNA expression profiles in regenerated scales were compared to those data collected in ontogenic scales to investigate the effects of salinity acclimation on skeletal growth and development. Gene ontology and KEGG enrichment analysis of the differentially expressed genes and differentially expressed miRNA target genes showed similar expression profiles. These genes were mainly related to ion and energy metabolism.

**Abstract:**

In order to explore the potential effects of salinity acclimation on bone metabolism of rainbow trout (*Oncorhynchus mykiss*), transcriptional information of regenerated scales under salinity acclimation (sea water, SW) was compared to those of fish under fresh water (FW) environments. According to the high-throughput sequencing results, a total of 2620 significantly differentially expressed genes (DEGs) were identified in the data of SW vs. FW. Compared with the FW group, six significantly downregulated and 44 significantly upregulated miRNAs were identified in the SW scales (*p* < 0.05). Furthermore, a total of 994 significantly differentially expressed target genes (DETGs) were identified from the 50 significantly differentially expressed miRNAs (DE miRNAs). Gene ontology analysis of the aforementioned DETGs was similar to the results of the differentially expressed genes (DEGs) obtained from mRNA-seq data, these genes were mainly related to ion metabolism. KEGG enrichment analysis of the DEGs and DETGs suggested that many significantly enriched pathways were related to the energy metabolism pathway.

## 1. Introduction

Fish bones consist of internal skeleton and exoskeleton. Processes and signaling pathways involved in matrix deposition, mineralization and matrix resorption are shared between bones and elasmoid scales. The easy access of scales makes it an ideal model for bone metabolism research. As part of the exoskeleton bony structure, scales are of mesodermal origin and have the capacity of regeneration when injured [1]. Elasmoid scales consist of mineralized collagen matrix and associated scleroblasts (scale-forming cells, analogous to osteoblasts) and scleroclasts (scale-resorbing cells, analogous to osteoclasts) [2]. When a scale is removed, lining cells that remain in the scale pocket differentiate to start forming a new one, fully regrown and mineralized within a matter of weeks [2]. Therefore, the generation paradigm of scales can provide researchers a model to study principles underlying skeletal growth and development in an adult fish.

Rainbow trout (*Oncorhynchus mykiss*) has long been considered one of the excellent breeding species. The adaptation of this species to salinity is important for developing cage culture in sea water [3]. Despite the efforts to understand how rainbow trout cope with different salinities (osmoregulation mechanism), not much consideration has been given to how the other systems (such as the skeletal system) might respond to the salinity adaptation. Recently, the effect of salinity acclimation on bone metabolism of *O. mykiss* was confirmed based on the histochemistry and morphometric analysis of the scales [4]. Considerable interest underlying the molecular mechanisms of scale regeneration is important to establish the salinity response of bone homeostasis in fish.

MicroRNA (miRNA) plays a switching or fine-tuning regulatory role according to the mRNA expression level of their target genes [5]. Here in this study, the global mRNA and miRNA profile of regenerated scales in rainbow trout under salinity acclimation were studied using the Illumina sequencing platform. The differentially expressed genes (DEGs) and differentially expressed miRNAs (DE miRNAs) were characterized. Comprehensive analysis of transcriptome data obtained in regenerated and ontogenic scales was performed to provide a more comprehensive understanding of the molecular response of bone homeostasis to salinity acclimation.

## 2. Materials and Methods

### 2.1. Sample Preparation

A total of 90 fish were equally separated into two groups named sea water (SW) and fresh water (FW). Each group had three replicates. Preparation and management of these fish were performed according to our related report [4]. At the start of the experiment, ontogenic scales of each fish were removed from a 2 cm × 1 cm area on the left flank of the fish (extending from the posterior base of the dorsal fin to the base of the caudal fin) after sedation with MS-222 (3-Aminobenzoic acid ethyl ester methanesulfonate). Afterwards, fish in the FW group remained in fresh water (3 ppt), while fish in the SW group were subjected to salinity acclimation according to our related report [4]. Regenerated scales were collected at the time points of 14 days (14D) since the start of the salinity acclimation.

### 2.2. RNA Extraction, Library Construction, mRNA and Small RNA Sequencing, and Analysis

The experiment and relative analysis were carried out according to our related report [4]. The raw data have been uploaded into the NCBI database Sequence Read Archive (SRA) and the SRR numbers are from SRR15559974 to SRR15559979. The context score of the predicted target genes by TargetScan (https://www.targetscan.org/vert_80/, accessed on September 2021) was set as equal or more than 50 and miRanda Energy was set as less than −10 [6]. Expressions of six randomly selected mRNAs and five miRNAs were detected using qRT-PCR and the primers are shown in Appendix A.

## 3. Results

### 3.1. General Description of the Small RNA-Seq Data and mRNA-Seq Data in Rainbow Trout

Based on High throughput sequencing, 9425934, 10669173, and 15870129 total raw reads were obtained from three scale samples from the SW group (named SW_1, SW_2, SW_3); 11000223, 10467693, and 11990645 total raw reads were obtained from three samples from the FW group (named FW_1, FW_2, FW_3). After filtering against mRNA, RFam, and Repbase databases, a minimum of 205759 and a maximum of 428760 valid uniq reads were obtained (Appendix A). In total, 1273 unique mature miRNAs originated from 1254 pre-miRNAs were identified; among them, 355 were novel pre-miRNA and 316 were novel unique mature miRNA (Appendix A).

The mRNA-seq of each scale sample yielded 77981856 to 90907282 raw reads. After filtering low quality reads, an average of 66724970 to 87003478 clean reads was obtained. More than 84.40% of clean reads were mapped to the *O. mykiss* reference genome (Appendix A).

### 3.2. Differentially Expressed miRNAs (DE miRNAs), Differentially Expressed Genes (DEGs), and Differentially Expressed Target Genes (DETGs) of the DE miRNAs in Regenerated Scales of SW Group

When compared with the FW group, expression of six miRNAs was significantly downregulated and expression of 44 miRNAs was significantly upregulated in the SW group (*p* < 0.05). Among these miRNAs, seven were identified as novel miRNAs, and expressions of 14 miRNAs were too low to be identified in the FW group. A miRNA family analysis was conducted for the 50 DE miRNAs; 20 of them were categorized into 16 miRNA families (Appendix A). As for mRNA-seq, a total of 2620 significantly differentially expressed genes (DEGs) were identified in regenerated scales of the SW group. The DE miRNAs were clustered based on the similarity of miRNA expression profiles. The results showed that six downregulated miRNAs could be clustered into two parts. Similarly, 44 unregulated miRNAs were significantly clustered into two clusters (Appendix A).

A total of 994 significantly differently expressed target genes were identified from the 50 DE miRNAs (*p* < 0.05); 334 of them were upregulated and 660 were downregulated in the SW group. 

The expression patterns of the five randomly selected miRNAs and six mRNAs were completely identical with the transcriptome sequencing results (Appendix A).

### 3.3. GO Functional Enrichment Analysis of DEGs and DETGs

GO enrichment analysis revealed that metal ion binding had the largest number of DEGs in the classification of molecular function; the cellular component had the highest number of DEGs enriched in the membrane and the integral component of the membrane; and the highest number of DEGs in the biological process were enriched in transport and oxidation-reduction process (Figure 1). Similar results were found in GO analysis of DETGs (Figure 2). In addition, GO analyses of DETGs of up- and down-regulated miRNAs in the SW group are, respectively, shown in Appendix A.

### 3.4. KEGG Pathway Annotation of DEGs and DETGs

The identified DEGs were significantly enriched in a total of 51 pathways (*p* < 0.05) (Appendix A); oxidative phosphorylation, citrate cycle (TCA cycle), hypertrophic cardiomyopathy (HCM), dilated cardiomyopathy (DCM), and propanoate metabolism were the top five significantly enriched pathways (Figure 3A). The DETGs were significantly enriched in a total of 41 pathways (*p* < 0.05) (Appendix A); these pathways mainly consisted of valine, leucine and isoleucine biosynthesis, thyroid cancer, hypertrophic cardiomyopathy (HCM), dilated cardiomyopathy (DCM), and 2-oxocarboxylic acid metabolism pathways (Figure 3B).

## 4. Discussion

Salinity change affects the ion transport and osmotic pressure of fish, and the required energy for osmo-regulation could be provided by the lipid. Therefore, not only osmotic pressure regulation, but also other processes such as lipid metabolism and the composition of fatty acids in tissues could be affected [7]. In rainbow trout, both osmo-regulation and energy metabolism were shown to be the core regulatory contents affected by salinity change [8]. In order to explore the effect of salinity acclimation on bone metabolism of rainbow trout, the global miRNA expression profiles in ontogenic scales were examined, and the DETGs were associated with terms such as metal ion binding, ATP binding, nucleotide binding and transferase activity; KEGG pathway analysis also revealed that they were mainly enriched in bone metabolism-related signaling pathways such as MAPK, calcium and Wnt [4]. However, the special regeneration capacity of scales has led us to another interesting question—whether the regeneration process would be affected by environmental factors such as salinity. Transcriptional information of regenerated scales was analyzed in this study, and the obtained DEGs in regenerated scales of the SW group were mainly enriched in biological processes such as transport and oxidation reduction. These DEGs mainly participated in the transport terms involving ion exchange and transport of chloride ions, sodium ions, and calcium ions; other functions were involved in the transport of substances such as amino acid and fatty acid transporter. Moreover, the DEGs and DETGs in other tissues of steelhead trout (*O. mykiss*) also showed similar results [9], indicating that many similar pathways such as ion and material transport and the regulation of cell membrane were activated in the different types of tissues. Meanwhile, KEGG enrichment analysis of DEGs and DETGs also exhibited a similar result in the significantly enriched pathways such as Valine, leucine, and isoleucine degradation, oxidative phosphorylation and citrate cycle (TCA cycle). Since the energy metabolism of fish could be affected by salinity stress [10,11], and the energy source mainly depended on oxidative phosphorylation [12], the aforementioned metabolic-related pathways were normally affected by salinity changes. The enrichment of DETGs in both ontogenic and regenerated scales of rainbow trout both revealed similar results, indicating that the regulatory process was basically consistent with the normal tissues.

The miRNAs play an important role in regulating physiological activities in rainbow trout [13]. To date, a large number of miRNAs have been confirmed to be extensively involved in various physiological activities in skeleton development of fish, such as miR-20a [14], miR-133, and miR-222 [15] for bone formation, and miR-25 for regulating osteoclasts [16]. In addition to the process of skeleton development, miRNAs in the miR-8, miR-10, and miR-30 family were found to be involved in the osmotic regulation [17,18].

Here in this study, classification of DE miRNAs indicated that not only the members in the miR-30 and miR-10 family, but also members in the miR-181 and miR-25 family were significantly changed in regenerated scales when rainbow trout were subjected to salinity acclimation. Similarly, expression of some members in the miR-25 and miR-181 family was also changed in ontogenic scales [4]. Possible roles of these miRNAs in bone metabolism when fish were subjected to salinity acclimation still need further research.

## Figures and Tables

**Figure 1 animals-12-01265-f001:**
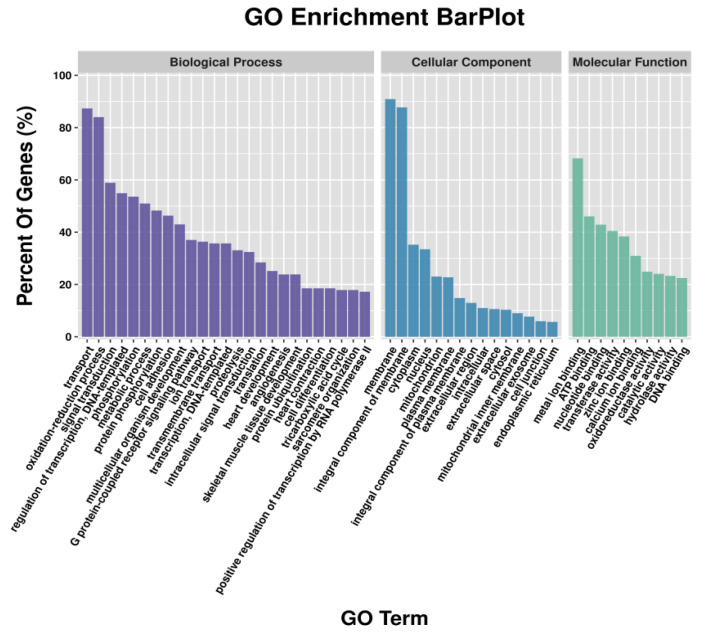
Functional classification of the differential expression genes (DEGs) by mRNA-seq according to GO category.

**Figure 2 animals-12-01265-f002:**
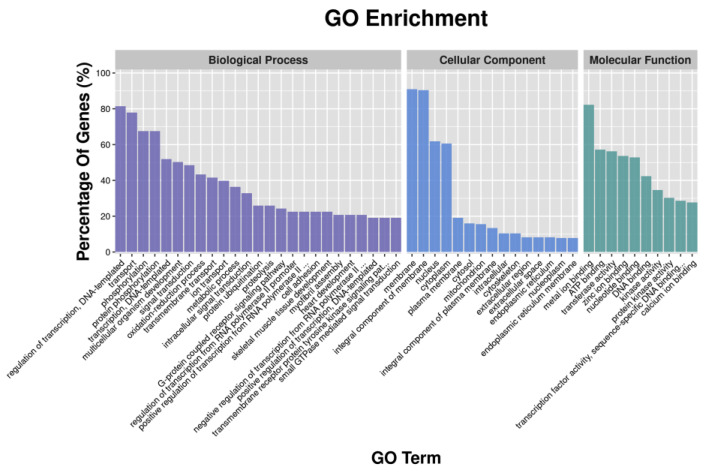
Functional classification of the differential expression target genes (DETGs) of differential expression miRNAs according to GO category.

**Figure 3 animals-12-01265-f003:**
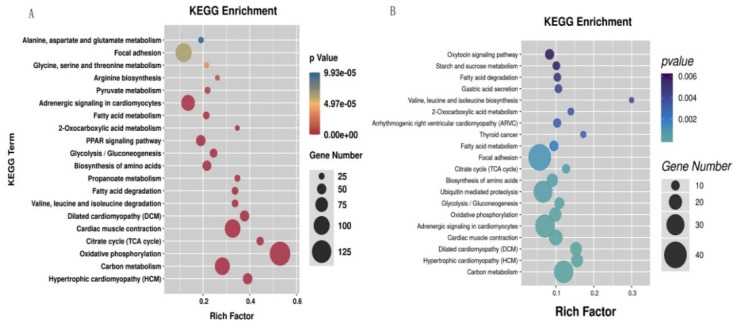
The top 20 significantly enriched pathways in KEGG enrichment of the differential expression genes (DEGs) (**A**) and the differential expression target genes (DETGs) (**B**).

## Data Availability

All data used in the current study are available from the corresponding author on reasonable request.

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
