# Peer review of "Differentially Expressed miRNAs and mRNAs in Regenerated Scales of Rainbow Trout (Oncorhynchus mykiss) under Salinity Acclimation"

_animals, 2022, doi:10.3390/ani12101265_

Round 1

Reviewer 1 Report

I found that the authors have revised the manuscript, although some minor revisions are still required before publication.

1) In Results, Sections 3.5 and 3.6 should be changed to Sections 3.3 and 3.4, respectively.
2) Figure 4, although cited in Section 3.6, does not appear in the manuscript.
3) In the authors reply to this reviewer, the authors mentioned “Identification of the target mRNAs of miRNAs was performed according to Nam J-W et al (2014)”. Thus, the reference (Nam J-W et al. Molecular Cell, 2014) should be cited in Materials and Methods.

Reviewer 2 Report

 In order to understand the mechanisms of scale regeneration in fish, The authors invetigated the expression of microRNA (miRNA) and messenger RNA (mRNA) comprehensively in rainbow trout skin after removing scales and acclimation by seawater. This manuscript show that gene related to metal ion and energy metabolism changed in seawater acclimated rainbow trout. However, I have several concerns that the author should deal.

Comments:

  1. The aim of this research is to understand the mechanisms of scale remodeling process by comprehensive analysis of gene expression. However, the author compared the gene expression only between freshwater rainbow trout and seawater acclimated fish, and they have not analyzed the effects of scale removing for gene expression in skin. Therefore, it is not clear what genes relate to scale remodeling in this experiment. The authors should investigate and discuss what miRNA and mRNA relate to scale remodeling. Furthermore, the authors had also compared the gene expression of miRNA and mRNA between freshwater fish and seawater acclimated fish in another paper “Effects of Salinity Acclimation on Histological Characteristics and miRNA Expression Profiles of Scales in Juvenile Rainbow Trout (Oncorhynchus Mykiss)”, which is under peer review of BMC Genetics. It is not clear the difference of research design and results between this paper and another paper of BMC. Hence, the authors should discuss the results of similarities and differences in detail between this paper and another paper of BMC.
  1. In this paper, the expression patterns of five randomly selected miRNA and six randomly selected mRNA were investigated by qPCR. However, it is not clear that the relationship between these miRNA and mRNA for scale remodeling and/or osmoregulation system in rainbow trout. Therefore, the authors should measure miRNA expression, and those target mRNA expression for understand the function of miRNA and their related mRNA on scale remodeling or osmoregulation.
  2. In all figures, the letters are too small to read (e. g. the GO Terms in Fig. 1 and 2, and KEGG terms in Fig. 3).
  3. L148-158. The author mentioned about transcriptome analysis in liver of rainbow trout citing reference no. 8. However, this reference used spotted sea bass for RNA-seq. The authors should confirm references.
  4. Correct “sig-nificant” to “significant”.
  5. Correct “ac-climation” to “acclimation”.

Round 2

Reviewer 2 Report

The authors answerd my comments and the second manuscript has been revised well. I think this manuscript will be acceptable. 

This manuscript is a resubmission of an earlier submission. The following is a list of the peer review reports and author responses from that submission.

Round 1

Reviewer 1 Report

In the present study, Yang et al. identified mRNAs and miRNAs expressed in rainbow trout scales regenerated under the conditions of seawater (SW) and freshwater (FW): a total of 2620 mRNAs and 50 miRNAs are differentially expressed. It would be possible that such differentially expressed miRNAs may play important roles, by repressing the target mRNA translation and/or inducing the target mRNA degradation in regenerated scales, in response to sea water. Potentially, the data may provide an important insight into skeletal regeneration or bone homeostasis in teleost fish.

My major concern is, although the FW samples are just a triplicate in the same experimental condition, there is a large difference in miRNA expression when comparing FW1 with FW2 and 3, with almost no miRNAs highly expressed in FW1 samples. Thus, the authors may need to re-examine the FW samples. Alternatively, at least, expression of 6 miRNAs at the bottom in Figure 2 should be evaluated by RT-qPCR.

Specific comments:

1) The first three letters of miRNA, gga, ssa, dre, and so on, represent the organisms (Gallus gallus, Atlantic salmon, Danio rerio, and so on). In the manuscript, the name of miRNA identified in rainbow trout should be indicated as miR-XXX-X or omy-miR-XXX-X. In addition, it is required to show a list of sequence information for 50 DE miRNAs in supplementary data.

2) Section 2.1: Do you have any reason why the regenerated scales were collected at 14 days?

3) Section 2.4: The authors need to explain in detail how the target mRNAs of miRNAs were identified. Only target sequences in the 3'UTR were identified? Please provide the URL for TargetScan 50.

4) Section 3.4: Differentially expressed target mRNAs should be analyzed separately for the increased miRNAs in SW and the decreased miRNAs in SW. GO analyses should be also performed separately between mRNA candidates targeted by the miRNAs that are increased in SW and those by the miRNAs that are reduced in SW. It would be important to show a list of identified DETGs as Supplementary data.

5) Some miRNAs identified in this study may be reported previously. In Discussion, it would be informative to cite previous reports that identified various miRNAs in rainbow trout (for example, Ramachandra er al, BMC Dev. Biol. 2008; Berthelot et al, Nature Comm.2014; Juanchich et al, BMC Genomics 2016).

Reviewer 2 Report

Paper rejected because plagiarism has been detected with a manuscript with a similar structure and content sent to BMC Genomics and being peer-reviewed now
